# A cascade of care analysis on the elimination of hepatitis C from public hospitals in Madrid

Jeffrey V. Lazarus [1,2 ✉], Marcela Villota-Rivas[1], Inmaculada Fernández[3], Francisco Gea[4], Pablo Ryan[5], Sonia Alonso López[6], Danielle Guy[1], José Luis Calleja[7,9] & Javier García-Samaniego[8,9]

## Abstract

**Background** Direct-acting antivirals can cure ≥95% of hepatitis C virus (HCV) cases, but do not reach everyone in need. This cross-sectional study analyses the HCV cascade of care (CoC) in Madrid, Spain, in high-risk patients, to inform micro-elimination measures.

**Methods** From September 2019 to May 2021, data from medical records were collected and analysed from six public hospitals in Madrid, including seven adult, high-risk patient groups: patients in haemodialysis or pre-dialysis programmes, co-infected with HIV, with advanced liver disease (ALD), with hereditary haematological diseases, with transplants and people who inject drugs (PWID).

**Results** Here we present an analysis of 3994 patients (68.8% male), 91.2% were tested for anti-HCV and 28.9% were positive. Of the total, 34.5% were tested for HCV–RNA and 62.4% of these were positive. Of those HCV–RNA positive, 98.0% were treatment-eligible: in 7.4%, treatment is ongoing and in 89.3% completed. Of the latter, 92.2% obtained a sustained virological response 12 weeks post treatment (SVR12). Of those with ongoing or completed treatment, 9.8% experienced loss to follow-up (LTFU) or had unknown SVR12, 50.3% developed hepatic and 20.3% extrahepatic complications. ALD patients had the highest proportion of HCV–RNA positives (32.5%). The lowest proportion of patients treated were PWID (85.2%).

**Conclusions** Almost one in ten high-risk patients in six of Madrid's public hospitals remains untested for HCV antibodies. An almost equal percentage of those untested have experienced LTFU, with the highest proportion in PWID. This approach to monitoring the HCV CoC is vital to inform measures to eliminate HCV in hospitals.

**Plain language summary**

Despite the existence of effective treatments with few side effects for hepatitis C virus (HCV), such treatments do not reach everyone in need and this means we cannot eliminate HCV. Here, we analysed HCV diagnoses, patients' access to care and treatment rates in high-risk populations in major public hospitals in Madrid. Data were collected from adult patients in haemodialysis or pre-dialysis programmes, co-infected with HIV, with advanced liver disease, with hereditary haematological diseases, with transplants and people who inject drugs (PWID). Nearly 10% of high-risk patients in six of Madrid's public hospitals did not have an initial test for HCV. An almost equal percentage of those who were not tested for HCV have not continued to be followed for care, primarily PWID. This approach to monitoring the HCV cascade of care is vital to inform measures to eliminate HCV in hospitals.

[1] Barcelona Institute for Global Health (ISGlobal), Hospital Clínic, University of Barcelona, Barcelona, Spain. [2] Faculty of Medicine, University of Barcelona, Barcelona, Spain. [3] Hepatology Unit, Hospital Universitario 12 de Octubre, Madrid, Spain. [4] Department of Hepatology, Hospital Universitario Ramón y Cajal, Madrid, Spain. [5] Hospital Universitario Infanta Leonor, CIBER en Enfermedades Infecciosas (CB21/13/00044), Madrid, Spain. [6] Hepatology Unit, Hospital General Universitario Gregorio Marañón, Madrid, Spain. [7] Department of Gastroenterology, Hospital Universitario Puerta de Hierro de Majadahonda, Madrid, Spain. [8] Hepatology Unit, Hospital Universitario La Paz, CIBERehd, IdiPAZ, Madrid, Spain. [9]These authors contributed equally: José Luis Calleja, Javier García-Samaniego. ✉email: Jeffrey.Lazarus@isglobal.org

An estimated 58 million people worldwide have chronic hepatitis C virus (HCV) infection[1], causing roughly 300,000 deaths annually, primarily from cirrhosis and liver cancer[2]. Direct-acting antiviral (DAA) therapy can cure ≥95% of HCV-positive people[3], which galvanised the World Health Organization (WHO) to call for the elimination of HCV as a public health threat by 2030[4]. Nonetheless, gaps in the HCV care continuum preclude treatment from reaching most people who need it[5]. Testing is crucial, as only 26% of the estimated global HCV-positive population know their status[1] and the virus often remains asymptomatic for years while progressively causing liver damage[2].

Although as of 2018, Spain was one of only 12 countries on track to eliminate HCV by 2030[6], current trends suggest that increased momentum is required to achieve this[7,8] and that there have been major setbacks during the COVID-19 pandemic[9]. The Spanish Association for the Study of the Liver (AEEH) estimates that there are 22,500 individuals with HCV unaware of their status in Spain[10]. Late diagnosis and presentation to care are major problems[10–14], with 28.1% found to be diagnosed late at hospitals[15] highlighting the importance of accurate epidemiological data. Late diagnosis and presentation to care may lead to a worse prognosis, reflecting the interplay of barriers hindering elimination efforts including a lack of public HCV awareness, ineffective systems directing HCV-diagnosed people to appropriate care pathways and stigma associated with behaviours that facilitate spreading HCV, such as injecting drug use[16].

These obstacles persist despite Spain's commitment to addressing HCV, demonstrated by the national government introducing a comprehensive HCV strategy in 2015[17], extending DAA treatment to all patients in 2017[18], and issuing testing guidance in 2020[19]. Sub-nationally, many regions/autonomous communities have proven their commitment to elimination, for example, La Rioja reported reaching the WHO target of diagnosing >90% of the population already in 2018[20], Cantabria has universal HCV screening and an elimination strategy[21], and Catalonia has implemented micro-elimination strategies that are supported by the health authorities[22]. However, such commitment is not uniform across Spain and country-wide HCV elimination requires optimal resource utilisation and multistakeholder commitment [4].

Micro-elimination can improve the effectiveness of national HCV-elimination efforts by breaking down prevention and treatment challenges into actionable tasks via the analysis of the situation in at-risk populations or subnational geographic settings[23]. This encourages stakeholder involvement, produces measurable, short-term results, often without large resource expenditures, and spurs the collection of high-quality clinical and epidemiological data from subpopulations of interest to design tailored strategies [24,25].

Large cities can serve as ideal environments to implement micro-elimination strategies due to the proximity of services relating to the HCV cascade of care (CoC) for diagnosis, linkage to care and treatment. This approach has been initiated by the Alliance for the Elimination of Viral Hepatitis in Spain in cities including Gijón, Granada, Santander, Sevilla and Valencia[26]. The micro-elimination approach and understanding the CoC can help improve data collection and communication amongst stakeholders to enhance care. The objective of this study was to analyse the HCV CoC in high-risk populations in major public hospitals in Spain's capital, Madrid, to provide accurate data to support multidisciplinary micro-elimination efforts.

In summary, we analysed HCV diagnoses, linkages to care and treatment rates in high-risk populations in major public hospitals in Madrid. We found that almost one in ten high-risk patients in six of Madrid's public hospitals did not have an initial test for HCV. An almost equal percentage of those who were not tested for HCV were lost to follow-up (LTFU).

## Methods

This is a cross-sectional observational study of patients receiving care through Madrid's public health services and who are at high risk of having HCV. From 1 September 2019 to 28 May 2021, anonymised data were collected in Microsoft Excel through a retrospective registry review of adult patients (18 years or older) in haemodialysis or pre-dialysis programmes, co-infected with HIV, with advanced liver disease (ALD), with hereditary haematological diseases (HHD), with transplants and of people who inject drugs (PWID) (see Supplementary Data). Six public hospitals in Madrid were included, covering a population of more than two million people: Hospital General Universitario Gregorio Marañón, Hospital Universitario Infanta Leonor, Hospital Universitario La Paz, Hospital Universitario Puerta de Hierro Majadahonda, Hospital Universitario Ramón y Cajal and Hospital Universitario 12 de Octubre, and associated addiction clinics. Data on PWID were obtained from the latter. Assuming a total population of 6000 patients (in all target groups) across all six hospitals, a sample size of 546 was needed in order to be 95% confident with a precision of +/−4 degrees. To reach this number, each centre was requested to provide data for at least 125 patients in each target population, which in theory would have yielded a sample size of 5250. Even though every centre was not able to provide data for 125 patients per target population group, we still had a sample size of 3994, which is greater than the required sample size ($n = 546$) to ensure generalisability of study findings.

Data in the registry were collected by authorised specialists working with these high-risk patients, including nephrologists (for haemodialysis, pre-dialysis and kidney-transplant patients); infectious-disease (ID) specialists (for patients co-infected with HIV); internal medicine, addiction specialists or ID doctors (for PWID seen in addiction centres linked to the hospitals); haematologists (for patients with HHD and bone marrow transplants); hepatologists (for ALD and liver-transplant patients); cardiologists (for cardiac-transplant patients); and pulmonologists (for lung-transplant patients). Anyone with an active HCV infection was treated in the hepatology or ID departments, and thus, all data pertaining to HCV care (i.e. post positive HCV–RNA diagnosis) were collected by the treating physicians in these departments. Data collected for the registry review included high-risk-group category/categories, date of birth, gender, country of origin, whether an HCV antibody (anti-HCV Ab) test was offered or not, anti-HCV Ab test result, whether an HCV–RNA test was offered or not, HCV–RNA test result, stage of liver fibrosis at diagnosis, whether the patient was eligible for treatment or not, stage of liver fibrosis pre-treatment initiation, treatment status, sustained virological response 12 weeks' post-treatment (SVR12) status, liver complications, extrahepatic complications, whether the patient had hepatocellular carcinoma (HCC) or not and whether the patient had decompensated cirrhosis or not. Registry-review data from each specialty were collected by a junior doctor and verified and managed by one focal researcher, a senior hepatologist or ID physician, per hospital, who was in charge of the process. These data were also reviewed by another researcher and clarifications were sought as needed; once verified, data were extracted per hospital and high-risk group, and analysed. A data quality check was performed by another researcher on 5% of the data and analyses of the data were run independently by two different researchers to ensure coherence.

**Variables**. Each high-risk-group category was number-coded and patients could fall into one or multiple categories. If a patient belonged to multiple categories, they were included and analysed in each one to account for the fact that they had more than one opportunity to be screened for HCV. The date of birth of patients was collected and used to calculate age. Gender was number-coded and could be "male", "female" or "trans". Country of origin was re-categorised into "Spanish" and "non-Spanish" and con-sisted of the response "Spain" versus all other possible nationalities. Tests offered status (anti-HCV Ab and HCV–RNA) and their results were number-coded and options included "no test offered" or "test offered" and "negative" or "positive", respectively. Liver-fibrosis stage was number-coded and options ranged from fibrosis stage 0 (F0) to 4 (F4) and all patients with fibrosis ≥F3 were categorised as having ALD as per the consensus definition of Mauss et al. for the late presentation of chronic viral hepatitis for medical care[12], in addition to any other patient that clinicians reported as having ALD as based on other criteria such as transient elastography score. Treatment status was number-coded and could be "no", "yes, ongoing" or "treatment completed". SVR12 status was number-coded and options included "SVR12 not reached", "yes, SVR12 achieved", "treatment ongoing" or "LTFU/SVR12 unknown". Liver complications were number-coded and could be "none", "jaundice", "hepatic encephalopathy", "ascites" and "variceal bleeding". Patients could have none, one or multiple liver complications. Extrahepatic complications were number-coded and options included "none", "cryoglobulinemia", "other vasculitis", "arthritis/arthralgia", "monoclonal gammopathy", "type 2 diabetes mellitus" (T2DM), "renal impairment", "mental health issues" and "other". Patients could have none, one or multiple extrahepatic complications. Eligibility for treatment, HCC and decompensated cirrhosis status was number-coded and could be "no" or "yes".

**Statistical analyses**. The number of patients in each high-risk-group category were totalled and proportions calculated over the total number of patients. Mean age and standard deviations (SDs) were calculated for each high-risk group and overall, for all patients and for those HCV–RNA positive. Gender and nationalities were totalled for all patients and for those HCV–RNA positive and proportions calculated over the total number of patients per group and overall. Test-offered (anti-HCV Ab and HCV–RNA) status and anti-HCV Ab result status were totalled and proportions calculated over the total number of patients per group and overall. Anti-HCV Ab result-status analysis was carried out in this way because in some instances, patients were coded as being anti-HCV Ab positive, despite not having been offered an anti-HCV Ab test during the dates considered in the study period, but positive anti-HCV Ab status was known from an encounter previous to these dates. HCV–RNA testing-status analysis was done in this way because in some occasions, patients had been offered an HCV–RNA test, despite not having had an anti-HCV Ab test offered or when anti-HCV Ab negative; the reasons for why this may have been done include having previous knowledge of a patient's anti-HCV Ab-positive status or requesting that both tests be run simultaneously when, for instance, there is high suspicion that a patient is HCV–RNA positive. For CoC drop-off calculations, HCV–RNA testing-status analysis was totalled and proportions calculated over the total number of patients who were anti-HCV Ab positive per group and overall. This was done in this way as typically only "those who are anti-HCV Ab positive would be tested for HCV–RNA. HCV-RNA test results were totalled and proportions calculated over the total number of patients offered the HCV–RNA test per group and overall.

Eligibility for treatment status was totalled and proportions calculated over the total number of HCV–RNA-positive patients per group and overall. Treatment status was totalled and proportions calculated over the total number of patients eligible for treatment per group and overall. SVR12 status was totalled and proportions calculated over the total number of patients with completed treatment per group and overall. Liver fibrosis stage at diagnosis was totalled and proportions calculated over the total number of HCV-RNA-positive patients per group and overall. Liver-fibrosis stage pre-treatment initiation was totalled and proportions calculated over the total number of patients eligible for treatment per group and overall. Instances of no data for liver-fibrosis stage at diagnosis and pre-treatment initiation were also totalled and proportions calculated over the total number of patients eligible for treatment per group and overall. Hepatic and extrahepatic complications were totalled for patients with ongoing and completed treatment and proportions calculated over the total number of patients with ongoing and completed treatment per group and overall. There were instances where patients had no data for categories other than liver-fibrosis stage, from the point of anti-HCV Ab test-offered status onwards, due to LTFU; consequently, these patients were coded as LTFU and analysed as such.

Fisher's exact test was used to analyse what socio-demographic and clinical characteristics were associated with LTFU during the overall CoC using Stata SE version 16.1. Covariates tested included age, gender, migrant status and high-risk group. No data were excluded from the analyses.

**Ethics approval and consent to participate**. For this study, Institutional Review Board approval was only required from one Spanish hospital. This study received ethical clearance in 2020 from the Ethical Committee of the Hospital Clínic, Barcelona, Spain (identification number: HCB/2020/1017), the hospital to which the principal investigator of this study is affiliated. This study conforms to international ethical standards, including the Declaration of Helsinki. All participants gave their written informed consent prior to the inclusion in the study.

## Results
**Overall analysis**. In total, there were 3994 patients across all high-risk groups (125 of whom had two or more comorbidities) and 68.8% were male, 87.2% of Spanish origin and the mean age was 57.6 years (SD: 14.8) (Table 1). Of the total, 91.2% were tested for anti-HCV Ab and of those tested, 28.9% were positive (Fig. 1, Table 2). Also, of the total, 34.5% were tested for HCV–RNA and 62.4% of these were positive.

Of those HCV–RNA positive, 71.0% were male, 91.5% of Spanish origin and the mean age was 57.9 years (SD: 11.1). Also, of those HCV–RNA positive, 61.4% had ALD, based on a fibrosis score of ≥F3 (Table 3), and 98.0% were eligible for treatment. Of those eligible for treatment, 61.9% had a fibrosis score of ≥F3 pre-treatment initiation, and in 7.4%, treatment was ongoing and in 89.3% completed. Of those who completed treatment, 92.2% obtained SVR12 and 50.3% developed hepatic and 20.3% extrahepatic complications (Table 4). The most common hepatic complication was decompensated cirrhosis (16.6%) and the least was jaundice (2.9%). The most common extrahepatic complication was T2DM (6.6%) and the least common were vasculitis other than cryoglobulinamia and monoclonal gammopathy (0.4% for both).

**Analysis per high-risk group**. The highest proportion of patients included in the study were transplant patients (22.2%), most of whom were male (68.2%) and of Spanish origin (95.1%), with a

**Table 1 Demographic characteristics of patients.**

| | Haemodialysis n (%) | | Pre-dialysis n (%) | | HIV n (%) | | PWID n (%) | | HHD n (%) | | ALD n (%) | | Transplant n (%) | | Total n (%) | |
|---|---|---|---|---|---|---|---|---|---|---|---|---|---|---|---|---|
| | All (n = 633) | HCV+ (n = 46) | All (n = 31) | HCV+ (n = 0) | All (n = 869) | HCV+ (n = 237) | All (n = 427) | HCV+ (n = 122) | All (n = 271) | HCV+ (n = 6) | All (n = 877) | HCV+ (n = 285) | All (n = 886) | HCV+ (n = 163) | All (n = 3,994) | HCV+ (n = 859) |
| Age, mean (SD) | 65.8 (15.9) | 62.2 (11.3) | 63.6 (15.3) | NA | 49.2 (11.8) | 54.0 (11.2) | 47.2 (10.2) | 49.0 (9.3) | 64.9 (17.9) | 75.5 (12.3) | 61.2 (12.4) | 61.2 (11.5) | 59.2 (12.5) | 62.8 (9.6) | 57.6 (14.8) | 57.9 (11.1) |
| Gender male | 411 (64.9) | 31 (67.4) | 22 (71.0) | 0 (0.0) | 671 (77.2) | 189 (79.7) | 331 (77.5) | 87 (71.3) | 139 (51.3) | 5 (83.3) | 570 (65.0) | 172 (60.4) | 604 (68.2) | 126 (77.3) | 2748 (68.8) | 610 (71.0) |
| Gender female | 222 (35.1) | 15 (32.6) | 9 (29.0) | 0 (0.0) | 194 (22.3) | 48 (20.3) | 96 (22.5) | 35 (28.7) | 132 (48.7) | 1 (16.7) | 307 (35.0) | 113 (39.6) | 282 (31.8) | 37 (22.7) | 1242 (31.1) | 249 (29.0) |
| Gender trans | 0 (0.0) | NA | 0 (0.0) | NA | 4 (0.5) | 0 (0.0) | 0 (0.0) | NA | 0 (0.0) | NA | 0 (0.0) | NA | 0 (0.0) | NA | 4 (0.1) | 0 (0.0) |
| Spanish origin | 558 (88.2) | 43 (93.5) | 28 (90.3) | NA | 631 (72.6) | 206 (86.9) | 355 (83.1) | 98 (80.3) | 243 (89.7) | 5 (83.3) | 823 (93.8) | 275 (96.5) | 843 (95.1) | 159 (97.5) | 3481 (87.2) | 786 (91.5) |
| Non-Spanish origin | 75 (11.8) | 3 (6.5) | 3 (9.7) | NA | 238 (27.4) | 31 (13.1) | 72 (16.9) | 24 (19.7) | 28 (10.3) | 1 (16.7) | 54 (6.2) | 10 (3.5) | 43 (4.9) | 4 (2.5) | 513 (12.8) | 73 (8.5) |

ALD advanced liver disease, HCV hepatitis C virus, HHD hereditary haematological diseases, NA not applicable, PWID people who inject drugs.

mean age of 59.2 (SD: 12.5). The least proportion of patients included were pre-dialysis patients (0.8%). Pre-dialysis patients had the highest proportion of anti-HCV Ab testing (100.0%) and PWID had the lowest (67.9%). The highest proportion of anti-HCV Ab positives was found in PWID (64.2%) and the least in pre-dialysis patients (3.2%). PWID also had the highest proportion of HCV–RNA testing (57.8%) and pre-dialysis patients the lowest (3.2%). ALD patients had the highest proportion of HCV–RNA positives, most of whom were male (60.4%) and of Spanish origin (96.5%), with a mean age of 61.2 (SD: 11.5), both for the group overall (32.5%) and for those tested for HCV–RNA (94.1%), and pre-dialysis ones the lowest for both categories (0.0% for both).

Upon diagnosis, the transplant group had the highest proportion of patients with ALD (89.6%), based on a fibrosis score of ≥F3, and PWID had the lowest (23.8%). The highest proportion of patients eligible for treatment was found in HIV and PWID patients (99.2% for both) and the least in HHD patients (50.0%). Pre-treatment initiation, the ALD group had the highest proportion of patients with a fibrosis score of ≥F3 (81.9%) and PWID had the least (23.1%). The HIV group had the highest proportion of patients with ongoing or completed treatment out of those being HCV–RNA positive (97.5%) and the HHD group had the lowest (50.0%).

Of those eligible for treatment, the HHD group had the highest proportion of those with ongoing or completed treatment (100%) and PWID had the lowest (85.2%). The transplant group had the highest proportion of patients achieving SVR12 out of those being eligible for treatment (98.7%) and PWIDs the least (40.5%). Out of those who completed treatment, the haemodialysis group had the highest proportion of patients reaching SVR12 (100.0%) and the HHD one the lowest (66.7%). Of those with ongoing or completed treatment, transplant patients had the highest proportion of hepatic complications across all categories, with decompensated cirrhosis being the most common (50.0%) and jaundice the least (7.1%), and HHD patients had the least (0.0%, across all categories).

### Analysis of CoC drop-off
*Overall.* Of the total 3994 patients in all high-risk groups, 350 (8.8%) were not tested for anti-HCV Ab. Of those anti-HCV Ab positive, 3.1% were not offered an HCV–RNA test. Of those HCV–RNA positive and eligible for treatment, 3.3% were not treated. Of those with ongoing or completed treatment, 9.8% experienced LTFU or had unknown SVR12. Overall, LTFU at any point in the CoC was associated with being male ($p < 0.000$), in the youngest-age cohort (18–30 years old, $p < 0.000$), of non-Spanish origin ($p < 0.000$), having HIV ($p < 0.000$) and being a PWID ($p < 0.000$).

Of the 125 patients with two or more comorbidities, 53 (42.4%) were not tested for anti-HCV Ab. Of those anti-HCV Ab positive, 11.9% were not offered an HCV-RNA test. Of those HCV–RNA positive and eligible for treatment, 1.9% were not treated. Of those with ongoing or completed treatment, 28.3% experienced LTFU or had unknown SVR12.

*Per high-risk group.* The PWID group had the highest proportion of patients not tested for anti-HCV Ab (32.1%) and the pre-dialysis one the least (0.0%). Of those anti-HCV Ab positive, the PWID group had the highest proportion of patients not tested for HCV–RNA (9.9%), followed by the HIV group (3.3%); all other groups had a 0.0% proportion in this category. Of those HCV–RNA positive and eligible for treatment, the PWID group had the highest proportion of patients not treated (14.8%) and the HHD group the least (0.0%). Of those treated, the PWID group

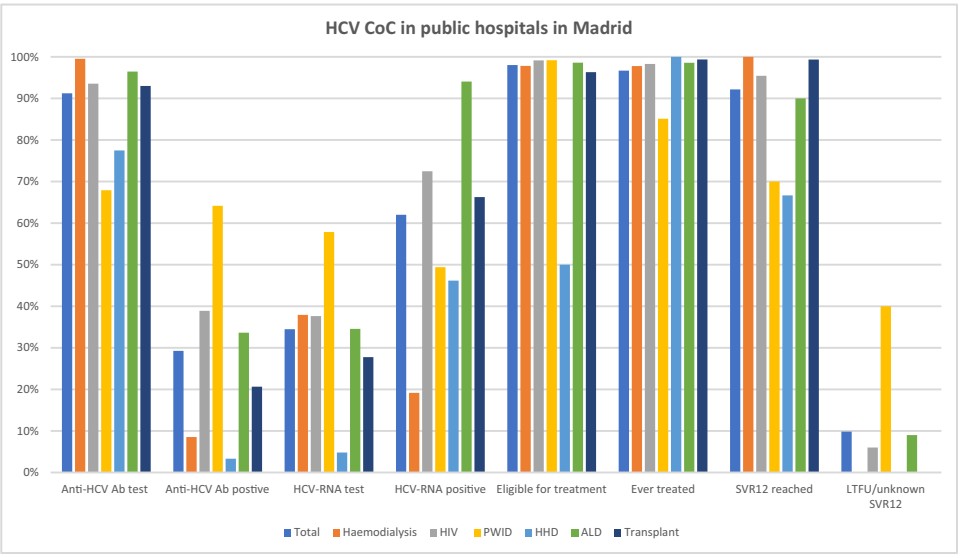

**Fig. 1 Analysis of the HCV CoC in public hospitals in Madrid.** Pre-dialysis patients were omitted, given the high proportion of anti-HCV Ab and HCV–RNA testing (100.0%), out of those anti-HCV Ab positive for the latter, and low proportion of HCV–RNA-positive patients (0.0%). Category 'Ever treated' includes patients with ongoing and completed treatment. Denominator for 'Anti-HCV Ab test', 'Anti-HCV Ab positive' and 'HCV-RNA test' is the total n of patients of each corresponding group; 'HCV-RNA positive' is the n of patients who were tested for HCV-RNA of each corresponding group; 'Eligible for treatment' is the n of patients who tested positive for HCV–RNA of each corresponding group; 'Ever treated' is the *n* of patients who were eligible for treatment of each corresponding group; 'SVR12 reached' is the n of patients who completed treatment of each corresponding group; 'LTFU/unknown SVR12' is the n of patients who were ever treated of each corresponding group. Anti-HCV Ab HCV antibody, ALD advanced liver disease, CoC cascade of care, HCV hepatitis C virus, HHD hereditary haematological diseases, LTFU lost to follow-up, PWID people who inject drugs, SVR12 sustained virological response 12 weeks post treatment.

had the highest proportion of patients experiencing LTFU or having an unknown SVR12 (39.8%); the haemolysis, HHD and transplant groups all had a 0.0% proportion in this category.

## Discussion

This cross-sectional study describes the HCV CoC for high-risk patients at six of Madrid's public health hospitals, covering at least one-third of the total population of the region. Overall, the hospitals involved in this study perform well in the initial steps of the CoC. Over 90% of those in high-risk groups were tested for HCV, which is considered the biggest challenge in achieving the WHO elimination goal[5]. This also aligns with the European Association for the Study of the Liver's HCV practice guidelines[27], Spanish national HCV strategy[17], HCV elimination-position statement of AEEH[10] and Madrid's HCV-elimination strategy[28], all of which guide HCV care in the centres participating in this study and emphasise HCV screening in high-risk groups. Furthermore, the vast majority of those eligible for treatment have either completed or are undergoing treatment. Given these results, the Madrid public hospital system outperforms the latest global and European estimates of the HCV CoC, where only 13% and 8% of those diagnosed have been successfully treated, respectively. The Madrid public hospital system also exceeds the WHO target of having 80% of individuals eligible for treatment treated by 2030, by having over 96% of individuals eligible for treatment either in ongoing treatment or treated[1]. As such, Madrid could serve as a valuable model of successful HCV care for other major European cities seeking to eliminate it by 2030.

Despite performing well overall, our findings also indicate gaps in the HCV CoC among high-risk patients in Madrid. Over 62% of the overall cohort tested positive for HCV–RNA and more than 61% of these individuals had ALD at diagnosis, highlighting the need for earlier diagnosis. Our estimates are in line with other research that highlights late diagnosis and presentation to specialist viral hepatitis care as an issue within the Spanish healthcare

system[10–15]. This is surprising given that there has been universal access to DAAs in Spain since 2017 and stresses the need for better screening efforts towards certain patient populations, such as transplant and ALD patients, who had the highest percentage of stages 3–4 liver fibrosis at diagnosis and pre-treatment initiation, respectively. PWID are also a very high-risk population group that require additional surveillance and linkage to care according to our screening and testing data. In addition to being one of the highest at risk groups for HCV globally[29] and late presentation to care in Spain[13], PWID in our study had the lowest proportion of individuals tested for anti-HCV Ab and treated for HCV infection and the highest rate of LTFU. This is in line with previous studies indicating poor HCV treatment uptake within this population[30]. This finding is unsurprising given the various structural, societal and health-system barriers that PWID face when trying to access the formal healthcare system. These include punitive drug laws, stigma by healthcare providers and HCV diagnostic testing requiring multiple visits[31]. As such, services at public hospitals will need to be complemented with community-based efforts. For example, point-of-care testing and peer-based models of care, which have been proven to improve testing and treatment rates among PWID, may be appropriate for HCV micro-elimination efforts within this population[31,32].

In addition to PWID, another key group that should be targeted for retention in the HCV CoC is migrants. Our results indicate that being a migrant was highly associated with falling off at any point in the CoC. This is in line with a similar study of the HCV CoC in Barcelona, where migrant PWID were more likely to experience LTFU in the CoC compared with Spanish-born PWID[33]. Previous evidence suggests that screening programes with culturally appropriate and relevant educational resources on HCV should be considered to reach this population[34]. This should be considered for Spain in particular, given that it has one of the largest migrant populations in Europe. Future research could also explore more deeply the reasons

**Table 2 Full analysis of the HCV CoC in public hospitals in Madrid.**

| | Haemodialysis n (%) | Pre-dialysis n (%) | HIV n (%) | PWID n (%) | HHD n (%) | ALD n (%) | Transplant n (%) | Total n (%) |
|---|---|---|---|---|---|---|---|---|
| Number of patients | 633 (15.8[a]) | 31 (0.8[a]) | 869 (21.8[a]) | 427 (10.7[a]) | 271 (6.8[a]) | 877 (22.0[a]) | 886 (22.2[a]) | 3994 (100.0[a]) |
| Anti-HCV Ab test performed | 630 (99.5[b]) | 31 (100.0[b]) | 813 (93.6[b]) | 290 (67.9[b]) | 210 (77.5[b]) | 846 (96.5[b]) | 824 (93.0[b]) | 3644 (91.2[b]) |
| Anti-HCV Ab positive | 54 (8.5[b]) | 1 (3.2[b]) | 338 (38.9[b]) | 274 (64.2[b]) | 9 (3.3[b]) | 295 (33.6[b]) | 183 (20.7[b]) | 1154 (28.9[b]) |
| HCV-RNA test performed | 240 (37.9[b]) | 1 (3.2[b]) | 327 (37.6[b]) | 247 (57.8[b]) | 13 (4.8[b]) | 303 (34.5[b]) | 246 (27.8[b]) | 1377 (34.5[b]) |
| HCV-RNA positive | 46 (19.2[c]) | 0 (0.0[c]) | 237 (72.5[c]) | 122 (49.4[c]) | 6 (46.2[c]) | 285 (94.1[c]) | 163 (66.3[c]) | 859 (62.4[c]) |
| Eligible for treatment | 45 (97.8[c]) | NA | 235 (99.2[c]) | 121 (99.2[c]) | 3 (50.0[c]) | 281 (98.6[c]) | 157 (96.3[c]) | 842 (98.0[c]) |
| Treatment ongoing | 0 (0.0[d]) | NA | 12 (5.1[d]) | 33 (27.3[d]) | 0 (0.0[d]) | 17 (6.0[d]) | 0 (0.0[d]) | 62 (7.4[d]) |
| Treatment completed | 44 (97.8[d]) | NA | 219 (93.2[d]) | 70 (57.9[d]) | 3 (100.0[d]) | 260 (92.5[d]) | 156 (99.4[d]) | 752 (89.3[d]) |
| SVR12 reached | 44 (100.0[c]) | NA | 209 (95.4[c]) | 49 (70.0[c]) | 2 (66.7[c]) | 234 (90.0[c]) | 155 (99.4[c]) | 693 (92.2[c]) |
| LTFU/unknown SVR12 | 0 (0.0[e]) | NA | 13 (5.6[e]) | 41 (39.8[e]) | 0 (0.0[e]) | 26 (9.4[e]) | 0 (0.0[e]) | 80 (9.8[e]) |

Total summation of percentages in 'Number of patients' row does not equal 100.0% due to rounding.
Anti-HCV Ab HCV antibody, ALD advanced liver disease, CoC cascade of care, HCV hepatitis C virus, HHD hereditary haematological diseases, NA not applicable, LTFU lost to follow-up, PWID people who inject drugs, SVR12 sustained virological response 12 weeks post-treatment.
[a]Percentage of the total n of patients.
[b]Percentage of the n of patients of the corresponding group/column.
[c]Percentage of the n of patients of the cell above.
[d]Percentage of the n of patients eligible for treatment of the corresponding group/column.
[e]Percentage of the sum of the n of patients with ongoing treatment and the n of patients with completed treatment.

**Table 3 Liver fibrosis stage at HCV diagnosis and pre-treatment initiation.**

| Liver fibrosis stage | Haemodialysis n (%) | | HIV n (%) | | PWID n (%) | | HHD n (%) | | ALD n (%) | | Transplant n (%) | | Total n (%) | |
|---|---|---|---|---|---|---|---|---|---|---|---|---|---|---|
| | Diagnosis (n = 46) | Pre-TI (n = 45) | Diagnosis (n = 237) | Pre-TI (n = 235) | Diagnosis (n = 122) | Pre-TI (n = 121) | Diagnosis (n = 6) | Pre-TI (n = 3) | Diagnosis (n = 285) | Pre-TI (n = 281) | Diagnosis (n = 163) | Pre-TI (n = 157) | Diagnosis (n = 859) | Pre-TI (n = 842) |
| F0 | 6 (13.0) | 5 (11.1) | 31 (13.1) | 30 (12.8) | 47 (38.5) | 45 (37.2) | 0 (0.0) | 0 (0.0) | 2 (0.7) | 1 (0.4) | 4 (2.5) | 20 (12.7) | 90 (10.5) | 101 (12.0) |
| F0-F1 | 0 (0.0) | 0 (0.0) | 0 (0.0) | 0 (0.0) | 2 (1.6) | 2 (1.7) | 0 (0.0) | 0 (0.0) | 0 (0.0) | 0 (0.0) | 0 (0.0) | 0 (0.0) | 2 (0.2) | 2 (0.2) |
| F1 | 12 (26.1) | 10 (22.2) | 60 (25.3) | 49 (20.9) | 22 (18.0) | 26 (21.5) | 0 (0.0) | 0 (0.0) | 31 (10.9) | 27 (9.6) | 3 (1.8) | 4 (2.5) | 128 (14.9) | 116 (13.8) |
| F2 | 6 (13.0) | 6 (13.3) | 33 (13.9) | 32 (13.6) | 6 (4.9) | 9 (7.4) | 1 (16.7) | 1 (33.3) | 22 (7.7) | 21 (7.5) | 8 (4.9) | 11 (7.0) | 76 (8.8) | 80 (9.5) |
| F3 | 7 (15.2) | 6 (13.3) | 28 (11.8) | 28 (11.9) | 10 (8.2) | 9 (7.4) | 1 (16.7) | 0 (0.0) | 36 (12.6) | 37 (13.2) | 13 (8.0) | 14 (8.9) | 95 (11.1) | 94 (11.0) |
| F4 | 12 (26.1) | 15 (33.3) | 75 (31.6) | 91 (38.7) | 19 (15.6) | 19 (15.7) | 3 (50.0) | 2 (66.7) | 190 (66.7) | 193 (68.7) | 133 (81.6) | 108 (68.8) | 432 (50.3) | 428 (50.9) |
| No data | 3 (6.5) | 3 (6.7) | 10 (4.2) | 5 (2.1) | 16 (13.1) | 11 (9.1) | 1 (16.7) | 0 (0.0) | 4 (1.4) | 2 (0.7) | 2 (1.2) | 0 (0.0) | 36 (4.2) | 21 (2.5) |

Total summation of percentages in some columns does not equal 100.0% due to rounding. ALD advanced liver disease, F0 fibrosis stage 0, F1 fibrosis stage 1, F2 fibrosis stage 2, F3 fibrosis stage 3, F4 fibrosis stage 4, HCV hepatitis C virus, HHD hereditary haematological diseases, PWID people who inject drugs, TI treatment initiation.

**Table 4 Analysis of hepatic and extrahepatic complications of patients treated for HCV.**

| Hepatic complications | Haemodialysis (n = 44) n (%) | HIV (n = 231) n (%) | PWID (n = 103) n (%) | HHD (n = 3) n (%) | ALD (n = 277) n (%) | Transplant (n = 156) n (%) | Total (n = 814) n (%) |
|---|---|---|---|---|---|---|---|
| Jaundice | 0 (0.0) | 7 (3.0) | 2 (1.9) | 0 (0.0) | 4 (1.4) | 11 (7.1) | 24 (2.9) |
| Hepatic encephalopathy | 2 (4.5) | 3 (1.3) | 0 (0.0) | 0 (0.0) | 8 (2.9) | 15 (9.6) | 28 (3.4) |
| Ascites | 6 (13.6) | 11 (4.8) | 2 (1.9) | 0 (0.0) | 21 (7.6) | 53 (34.0) | 93 (11.4) |
| Variceal bleeding | 2 (4.5) | 4 (1.7) | 0 (0.0) | 0 (0.0) | 10 (3.6) | 20 (12.8) | 36 (4.4) |
| HCC | 2 (4.5) | 7 (3.0) | 0 (0.0) | 0 (0.0) | 19 (6.9) | 68 (43.6) | 95 (11.6) |
| Decompensated cirrhosis | 5 (11.4) | 10 (4.3) | 1 (1.0) | 0 (0.0) | 44 (15.9) | 78 (50.0) | 135 (16.6) |
| Extrahepatic complications Cryoglobulinemia | 1 (2.3) | 11 (4.8) | 1 (1.0) | 0 (0.0) | 5 (1.8) | 1 (0.6) | 19 (2.3) |
| Other vasculitis | 0 (0.0) | 1 (0.4) | 0 (0.0) | 0 (0.0) | 0 (0.0) | 2 (1.3) | 3 (0.4) |
| Arthritis/arthralgia | 2 (4.5) | 1 (0.4) | 0 (0.0) | 0 (0.0) | 1 (0.4) | 0 (0.0) | 4 (0.5) |
| Monoclonal gammopathy | 1 (2.3) | 0 (0.0) | 0 (0.0) | 0 (0.0) | 1 (0.4) | 1 (0.6) | 3 (0.4) |
| T2DM | 5 (11.4) | 6 (2.6) | 2 (1.9) | 0 (0.0) | 24 (8.7) | 17 (10.9) | 54 (6.6) |
| Renal impairment | 13 (29.5) | 3 (1.3) | 0 (0.0) | 0 (0.0) | 0 (0.0) | 10 (6.4) | 26 (3.2) |
| Mental health issues | 0 (0.0) | 10 (4.3) | 2 (1.9) | 0 (0.0) | 13 (4.7) | 1 (0.6) | 26 (3.2) |
| Other | 2 (4.5) | 14 (6.1) | 1 (1.0) | 0 (0.0) | 10 (3.6) | 3 (1.9) | 30 (3.7) |

ALD advanced liver disease, HCC hepatocellular carcinoma, HCV hepatitis C virus, HHD hereditary haematological diseases, PWID people who inject drugs, T2DM type 2 diabetes mellitus.

for this LTFU by, for instance, tracking down and interviewing people who have experienced falling off the CoC.

Our results also highlight the importance of integrated care for high-risk groups. HCV often overlaps with other chronic conditions, such as HIV. Our results indicate that although people living with HIV had the highest HCV treatment ongoing and completed rates out of those HCV–RNA positive, having HIV was also associated with falling off at some point earlier in the CoC. This finding is surprising, given that HIV patients are typically on antiretroviral treatment and thus in regular care. However, this demonstrates the need for coordination among specialists, which can help link more people to care[5]. This is also evidenced by the fact that over 40% of people with two or more comorbidities were not screened for HCV and that almost a third experienced LTFU or had unknown SVR12. Better coordination may help detect and address health complications earlier, which may improve treatment outcomes.

**Strengths and limitations**. This is the first study to describe the HCV CoC in a large, urban setting in Spain. As such, it provides necessary evidence to monitor the advancement of targets to eliminate HCV as a public health threat in Spain and in Madrid more specifically. A particular strength of this study is that because it is a registry review that includes a heterogeneous set of study participants, external validity is strong. This is, however, a cross-sectional, observational study and therefore results do not infer causality. Furthermore, the CoC may vary over time. This registry review may have also been affected by some selection bias, as we only explored the HCV CoC in seven population sub-groups and there may be undetected and/or untreated HCV in other populations in Madrid's public hospitals. Therefore, it may not accurately represent the entire population of individuals with HCV in Madrid. Information bias was sought to be minimised by having one researcher reviewing all of the data and seeking clarifications as needed and, once verified, having another researcher perform a data quality check on 5% of the data and running analyses of the data independently by two different researchers to ensure coherence.

**Conclusions**
Public hospitals in Madrid perform relatively well in treating HCV. Notably, the vast majority of those eligible for treatment have either completed or are undergoing treatment. However, there is still room for improvement in diagnosing as even though most participants were screened for HCV, a considerable proportion tested positive for active infection and had ALD at diagnosis and pre-treatment initiation. Moreover, nearly 10% of high-risk individuals were not tested for HCV, despite almost a third of those being tested being anti-HCV Ab positive. Furthermore, an almost equal percentage of those untested for anti-HCV Ab have been LTFU. Overall, our findings underscore the need for hospitals and clinics across Spain and around the world to adopt an HCV CoC approach for all at-risk populations in order to inform elimination efforts. In the case of Madrid, the findings call for strengthening testing and linkage to care efforts for particularly vulnerable and high-risk groups, such as PWID, to reach the WHO goal of eliminating HCV as a public health threat within this decade.

**Reporting summary**. Further information on research design is available in the Nature Research Reporting Summary linked to this article.

**Data availability**
Source data for all analyses are available as Supplementary Data. Identifying information was removed from this dataset. To preserve patient anonymity, the raw dataset for this study cannot be made publicly available. However, it can be made available with appropriate ethical approval and by contacting the corresponding author.

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

## Acknowledgements

This study was funded by Gilead Sciences, Spain. J.V.L., M.V.-R. and D.G. acknowledge support to ISGlobal from the Spanish Ministry of Science, Innovation and Universities through the "Centro de Excelencia Severo Ochoa 2019–2023" Programme (CEX2018-000806-S) and from the Government of Catalonia through the CERCA Programme. D.G. is supported by a fellowship (LCF/BQ/DI20/11780008) from the "la Caixa" Foundation (ID 100010434). The authors would like to acknowledge the data quality check performed by Silvia Gómez-Araujo, ISGlobal.

## Author contributions

J.V.L. conceived of the study and developed the data-collection tool with J.G.-S. and J.L.C. I.F., F.G., P.R., S.A.L., J.L.C. and J.G.-S. led the data collection and M.V.-R. led the data management. Data analysis was undertaken by M.V.-R., D.G. and J.V.L. M.V.-R. and D.G. drafted the first iteration of the paper with J.V.L. All authors reviewed the full draft of the article, subsequent revisions and approved the final version for submission, including the authorship list.

## Competing interests

The authors declare no competing interests.
