## [Peer Review File · Communications Medicine]

Reviewers' comments:

Reviewer #1 (Remarks to the Author):

This is an informative paper describing 'cascade of care' (CoC) outcomes in several higher risk populations in Madrid hospitals. The premise of the paper is important: to document progress in addressing HCV infection given WHO goals of elimination as a public health imperative. The paper provides important view of potential gaps in care and opportunities for intervention. The study population and setting is relevant as the hospital setting is an important place to screen, diagnose and initiate treatment. Overall the methods are well described. I appreciate the well described process of data quality review. The following specific comments if addressed have potential to improve the paper.

1. Abstract: no comments

2. Introduction: relevant to a comment later in the Discussion, if these hospitals or the Ministry of Health in Madrid has a specific model or guidance, it would be good to state that here.

3. Methods:

- page 8: The supp. table is very informative and if the journal allows would be helpful to include in the full article.

-the FIGURE should better reflect the numbers in the table. I think that the figure does not accurately reflect forward movement on the CoC. This figure has 'running denominator - so the denominator for the bar in sVR treated is from the preceeding group (ever treated). Too really examine forward movement in the CoC and missed opportunities for care, a fixed denominator should be used. At the very least using the N that are RNA positive. Using a fixed denominator will be much more informative about lost opportunities with these patients.

4. Discussion

- Page 11, line 239. The authors haven't said what this Madrid 'model' is. Is there a policy or guideline document that can be referenced?

- Page 11, line 245. Why is this surprising? Is there an active surveillance system? Is liver disease not monitored for in older patients? this statement seems to suggest that more liver disease monitoring is needed in PWID - however - they did not have the highest proportion of liver fibrosis. I think this section needs a bit more of a 'microscope' to reflect what the data show.

- Page 11, line 246. This recommendation to better focus on PWID is certainly evident in the screening and testing data - this is your highest risk population but has a very low test uptake! Emphasizing this would improve this section.

Reviewer #2 (Remarks to the Author):

In this analyses the authors pretend to describe a cascade of care of HCV in 6 public hospitals in Madrid. In my opinion they do not present a cascade of care, since only individuals still in care were described, but a cross-sectional observational study to assess whether high risk individuals for having HCV were routinely tested and subsequently treated. They show that they have been quit succesfull in doing this routinely. Ie the title and methodology to me more fits a microelimination strategy in a public hospital setting aimed at high risk individuals.

The methodology used however remains unclear and should be clarified. As stated it seems to be a cross-sectional micro-elimination program, for which their own phycicians collected data but several

times in the manuscript it seems that tests were actively ordered for which participants gave their informed consent (as stated in the methods). In other parts the results seem more descriptive and also the discussion suggests a descriptive nature of their work/ retrospective data collection. This should be clarified in the methods and reflected in the title.

In the method section a power calculation is presented. To me it is completely unclear why this presented given the observational nature of their work.

Figure 1 describes their key findings. For clarity I would suggest to cluster the different risk groups instead of different steps.

Data was collected by the patients own phycians. Was any form of data verification performed? I assume patients were treated at the hepatology and/or ID departments for HCV. Was treatment and fibrosis data also primary collected by their own phycian or by the hepatologist/ID phycian? Were all individuals in care from the different risk groups included? And if yes how was this verified?

Reviewer #3 (Remarks to the Author):

The messages here are potentially important ones in the elimination effort for HCV but I am uncertain over a few areas. The major one is how the cohorts were compiled and how complete. I would accept that haemodialysis is a pretty concrete cohort but PWID in a secondary care sector? How do you know and what were they actually in hospital with? Without certainty over the denominator it is difficult to interpret the proportion positive. I would wish to see a more detailed explanation of how cohorts were accumulated and what safeguards there were in place to ensure that whatever search criteria used were universally employed.

Given this cohort is 2019 to 2021 I find it extraordinary that some well documented high risk groups (haemodiaysis, transplant and HIV positive patients) have such high positive rates. In UK practice these were almost the first micro-eliination cohorts and there will no longer be a single eligible HCV positive dialysis patient who is untreated and this has been the case for a number of years. Given policy in Spain is no different it is clear that some additional explanation is needed here and possibly a major call for change.

Dear Reviewers,

Thank you for reviewing our Original Article entitled "The elimination of hepatitis C virus infection from public hospitals in Madrid, Spain: A cascade of care analysis".

Your feedback and that of the reviewers helped us to improve the manuscript greatly. Find below the point-by-point answers to each of your suggestions.

Reviewer #1: This is an informative paper describing 'cascade of care' (CoC) outcomes in several higher risk populations in Madrid hospitals. The premise of the paper is important: to document progress in addressing HCV infection given WHO goals of elimination as a public health imperative. The paper provides important view of potential gaps in care and opportunities for intervention. The study population and setting is relevant as the hospital setting is an important place to screen, diagnose and initiate treatment. Overall the methods are well described. I appreciate the well described process of data quality review. The following specific comments if addressed have potential to improve the paper.

1. Abstract: no comments

2. Introduction: relevant to a comment later in the Discussion, if these hospitals or the Ministry of Health in Madrid has a specific model or guidance, it would be good to state that here.

RESPONSE: Thank you for your input. We have now specified in the discussion that the centres participating in this study follow the European Association for the Study of the Liver's HCV practice guidelines, Spanish national HCV strategy, HCV elimination position statement of the Spanish Association for the Study of the Liver and Madrid's HCV elimination strategy to guide HCV care and some detail about these (p 11).

3. Methods:

- page 8: The supp. table is very informative and if the journal allows would be helpful to include in the full article.

RESPONSE: Thank you for your input. The journal did indeed allow for the inclusion of this table in the article. We have included it as 'Table 2' in the manuscript (p 8, 20).

-the FIGURE should better reflect the numbers in the table. I think that the figure does not accurately reflect forward movement on the CoC. This figure has 'running denominator - so the denominator for the bar in sVR treated is from the preceding group (ever treated). Too really examine forward movement in the CoC and missed opportunities for care, a fixed denominator should be used. At the very least using the N that are RNA positive. Using a fixed denominator will be much more informative about lost opportunities with these patients.

RESPONSE: Thank you for your comment. We see your point in terms of how visually the figure may not seem reflective of the forward movement of the cascade of care (CoC). That being said, the figure is indeed reflective of Table 2 and the CoC and we thus propose a different approach from what you suggested to clarify and improve the figure, so that it becomes more clearly reflective of both the table and the CoC. Instead of changing any values, we have added notes under the figure (similar to the ones that we already had under Table 2) to clarify what the

denominator is for each point in the CoC (p 23). For instance, for the point you mentioned, 'SVR12 reached', the denominator is equal to those who completed treatment for each corresponding group. We think that using 'HCV-RNA positive' as a denominator instead would be misleading in terms of lost opportunities, since not all HCV-RNA positive patients were eligible for treatment and thus treated. We have also added the categories 'Eligible for treatment' and 'LTFU/unknown SVR12' to the figure, to help in improving the flow of the figure, while demonstrating lost opportunities in a more clear and visual format and staying reflective of the table and the CoC (p 23). Furthermore, we have also explained in the notes under the figure that 'Ever treated' includes those with ongoing and completed treatment, to improve clarity (p 23).

4. Discussion

- Page 11, line 239. The authors haven't said what this Madrid 'model' is. Is there a policy or guideline document that can be referenced?

RESPONSE: Thank you for your comment. As stated above, we have now specified in the discussion that the centres participating in this study follow the European Association for the Study of the Liver's HCV practice guidelines, Spanish national HCV strategy, HCV elimination position statement of the Spanish Association for the Study of the Liver and Madrid's HCV elimination strategy to guide HCV care some detail about them (p 11). We have also edited the sentence where we mention that due to its success, Madrid could serve as a model to follow for other European cities for HCV elimination, to increase its clarity (p 12).

- Page 11, line 245. Why is this surprising? Is there an active surveillance system? is liver disease not monitored for in older patients? this statement seems to suggest that more liver disease monitoring is needed in PWID - however - they did not have the highest proportion of liver fibrosis. I think this section needs a bit more of a 'microscope' to reflect what the data show.

RESPONSE: Thank you for this comment. We had already stated that this finding is surprising given that access to DAAs have been universal in Spain since 2017 but have now also edited the text to note that transplant and ALD patients should be better monitored as our findings show they had the highest rates of stages 3-4 fibrosis at diagnosis and pre-treatment initiation, respectively (p 12).

- Page 11, line 246. This recommendation to better focus on PWID is certainly evident in the screening and testing data - this is your highest risk population but has a very low test uptake! Emphasizing this would improve this section.

RESPONSE: Thank you for your suggestion. We edited the text around PWID by adding that they, "are also a very high-risk population group that require additional surveillance and linkage to care according to our screening and testing data. In addition to being one of the highest at-risk group for HCV globally¹⁴ and late presentation to care in Spain¹⁵,..." (p 12).

We also wanted the reviewer to know that the data collection ending period was edited from March to May 2021 as we decided, while working on the peer review responses, that even though all raw data came in by March of 2021, during the data verification process, which went on until May, data were edited and so we wanted the time period stated to be reflective of this.

Reviewer #2: In this analyses the authors pretend to describe a cascade of care of HCV in 6 public hospitals in Madrid. In my opinion they do not present a cascade of care, since only individuals still in care were described, but a cross-sectional observational study to assess whether high risk individuals for having HCV were routinely tested and subsequently treated. They show that they have been quit succesfull in doing this routinely. Ie the title and methodology to me more fits a microelimination strategy in a public hospital setting aimed at high risk individuals.

The methodology used however remains unclear and should be clarified. As stated it seems to be a cross-sectional micro-elimination program, for which their own physicians collected data but several times in the manuscript it seems that tests were actively ordered for which participants gave their informed consent (as stated in the methods). In other parts the results seem more descriptive and also the discussion suggests a descriptive nature of their work/ retrospective data collection.

This should be clarified in the methods and reflected in the title.

RESPONSE: Thank you for your comment. We agree that this is a cross-sectional observational study (the word “observational” was added to the methods in p 4, after “cross-sectional” to ensure clarity) and we also think that this work is indeed reflective of a cascade of care analysis, since by definition this process involves evaluating patient retention across the sequential care stages needed to attain a successful treatment outcome, which is what we have done. Furthermore, we do not agree that this fits a micro-elimination programme, since it is a retrospective registry review. We have added the word “retrospective” before “registry review” in the methods section (p 4) to ensure clarity, as per your suggestion. In terms of the tests that were ordered, this was all done previous to data collection, and we have edited the text in p 7 to help clarify this.

In the method section a power calculation is presented. To me it is completely unclear why this presented given the observational nature of their work.

RESPONSE: The power calculation is needed to ensure the generalisability of the study findings. We have added some text on p 5 to help clarify this.

Figure 1 describes their key findings. For clarity I would suggest to cluster the different risk groups instead of different steps.

RESPONSE: Thank you for your comment. Since this is a cascade of care (CoC) analysis, we clustered each sequential step in the CoC per high-risk group (except pre-dialysis patients for the reasons explained under the figure), in order to visually reflect the flow of the CoC. As per reviewer #1’s suggestion, we have improved the figure by adding the categories ‘Eligible for treatment’ and ‘LTFU/unknown SVR12’, to help in improving the flow and its clarity by explaining what the denominator is for each point in the CoC and that the category ‘Ever treated’ includes those with ongoing and completed treatment (p 23).

Data was collected by the patients own physicians. Was any form of data verification performed?

RESPONSE: Thank you for your question. As we have now clarified in the methods (p 4), as above, this was a retrospective registry review and as such we have edited the writing with regards to the

data collection process on p 5-6, to make things clearer. In terms of the data verification process, we have added and edited some text on p 6 to clarify how this was done.

I assume patients were treated at the hepatology and/or ID departments for HCV. Was treatment and fibrosis data also primarily collected by their own physician or by the hepatologist/ID physician?

RESPONSE: Thank you for your question. Your assumption is correct and patients were indeed treated for HCV by the hepatology/ID departments. Treatment and fibrosis data in the registry was collected by the treating physician (i.e., in the hepatology/ID departments). We have added a sentence on this in p 5 to ensure that this is clear.

Were all individuals in care from the different risk groups included? And if yes how was this verified?

RESPONSE: Thank you for your question. We have added some text to the methods section (p 5) to clarify that not all individuals from the different risk groups were included. As explained by the added text, we asked each hospital to provide at least 125 patients per risk group and even when they were unable to reach this number, we were still able to obtain a sample size much larger (n=3,994) than the required sample size (n=546) to ensure generalisability of study findings.

We also wanted the reviewer to know that the data collection ending period was edited from March to May 2021 as we decided, while working on the peer review responses, that even though all raw data came in by March of 2021, during the data verification process, which went on until May, data were edited and so we wanted the time period stated to be reflective of this.

Reviewer #3: The messages here are potentially important ones in the elimination effort for HCV but I am uncertain over a few areas. The major one is how the cohorts were compiled and how complete. I would accept that haemodialysis is a pretty concrete cohort but PWID in a secondary care sector? How do you know and what were they actually in hospital with? Without certainty over the denominator it is difficult to interpret the proportion positive. I would wish to see a more detailed explanation of how cohorts were accumulated and what safeguards there were in place to ensure that whatever search criteria used were universally employed.

RESPONSE: Thank you for your input. We have added some text in the methods section (p 5) to clarify that we asked each hospital to provide at least 125 patients per risk group to reach a sample size large enough to ensure generalisability of study findings and that even when they were unable to do so, we were still able to obtain a sample size much larger (n=3,994) than the required sample size (n=546) for generalisability purposes.

In terms of your point about PWID, we had already stated in the methods section that we worked with addiction clinics associated to the participating hospitals and have now added that it is from these clinics that we obtained the data on PWID (p 5), to improve clarity around this.

As for the safeguards used to ensure universal application of search criteria, as already mentioned on the methods section (p 4), data collection had a set date range (i.e., from September 2019 to May 2021—we edited the latter from March, please see note* below for an explanation) and we have added the exact dates (i.e., 1 September 2019 to 28 May 2021) to improve clarity (p 4). As for patient inclusion criteria, we had already stated and have further edited the text in the methods

section (p 4) that this was a “registry review of adult patients (18 years or older) in haemodialysis or pre-dialysis programmes, co-infected with HIV, with advanced liver disease (ALD), with hereditary haematological diseases (HHD), and with transplants, and of people who inject drugs (PWID)” to improve clarity. In terms of the data verification process, we added and edited some text on p 6 to clarify how this was done.

* The data collection ending period was edited from March to May 2021 as we decided, while working on the peer review responses, that even though all raw data came in by March of 2021, during the data verification process, which went on until May, data were edited and so we wanted the time period stated to be reflective of this.

Given this cohort is 2019 to 2021 I find it extraordinary that some well documented high risk groups (haemodialysis, transplant and HIV positive patients) have such high positive rates. In UK practice these were almost the first micro-elimination cohorts and there will no longer be a single eligible HCV positive dialysis patient who is untreated and this has been the case for a number of years. Given policy in Spain is no different it is clear that some additional explanation is needed here and possibly a major call for change.

RESPONSE: Thank you for highlighting this point. As per your suggestion, we have now noted the finding that a large portion of the cohort tested positive for HCV in the discussion (p 12) given that, as we had already stated on p 12, there has been prioritisation of micro-elimination efforts in these high-risk populations since 2017. This goes along with the text we already had in this section and edited slightly, about the ongoing late diagnosis and presentation to care issue in Spain, to explain this and how this “...stresses the need for better screening efforts towards certain patient populations...” like transplant patients (p 12), a major call for change.

REVIEWERS' COMMENTS:

Reviewer #1 (Remarks to the Author):

Dear Editor,

I have reviewed the authors' responses and revised manuscript. I think they did a very comprehensive revision and recommend accepting for publication.

Thank you for the opportunity to review this.

Kimberly Page

Reviewer #2 (Remarks to the Author):

i believe the manuscript has gained a lot of clarity by the adjustments made and have no further questions

Reviewer #3 (Remarks to the Author):

The authors have answered the points raised.

Dear Editors,

Thank you for reviewing our Original Article entitled "The elimination of hepatitis C virus infection from public hospitals in Madrid, Spain: A cascade of care analysis".

Your feedback and that of the reviewers helped us to improve the manuscript greatly. Find below the point-by-point answers to each of your suggestions.

Ref: COMMSMED-21-0454

Title: The elimination of hepatitis C virus infection from public hospitals in Madrid, Spain: A cascade of care analysis

Article Type: Original Article

Communications Medicine

Dear Professor Lazarus,

Your manuscript entitled "The elimination of hepatitis C virus infection from public hospitals in Madrid, Spain: A cascade of care analysis" has now been seen by 3 referees. You will see from their comments below that while they find your work of interest, some important points are raised. We are interested in the possibility of publishing your study in *Communications Medicine*, but would like to consider your response to these concerns in the form of a revised manuscript before we make a final decision on publication.

Please address the reviewers' concerns in full. Please note that we would encourage you to include the Supplementary Table within the main manuscript, according to Reviewer 1's suggestion.

We therefore invite you to revise and resubmit your manuscript, taking into account the points raised. Please highlight all changes in the manuscript text file.

We are committed to providing a fair and constructive peer-review process. Do not hesitate to contact us if you wish to discuss the revision in more detail or if there are specific requests from the reviewers that you believe are technically impossible or unlikely to yield a meaningful outcome.

At the same time, we ask that you ensure your manuscript complies with our editorial policies. Please see our revision file checklist for guidance on formatting the manuscript and complying with our policies. You will also find guidelines for replying to the referees' comments.

Communications Medicine seeks to improve the standards and transparency of reporting in our papers, and to ensure that all submissions conform with the editorial policies of Nature Research. When uploading your revised files please complete and submit Reporting Summary and Editorial Policy checklists as 'checklist' file types. Please note that these forms are a dynamic 'smart pdf' and must therefore be downloaded and completed in Adobe Reader, instead of being opened in a web browser. All points on the checklists must be addressed; if needed, please revise your manuscript in response to these points.

Your revised paper will not be returned to the editors for evaluation until these forms are provided.

Please use the following link to submit your revised manuscript, point-by-point response to the referees' comments (which should be in a separate document to the cover letter), reporting summary, editorial policy checklist and any additional files:

<https://mts-commsmed.nature.com/cgi-bin/main.plex?el=A6DK6Jf6A7fdo3I6A9ftdfL9ZHNAInI3YShcM8EHxgZ>

When submitting the revised version of your manuscript, please pay close attention to our Digital Image Integrity Guidelines and to the following points below:

- that unprocessed scans are clearly labelled and match the gels and western blots presented in figures.
 - that control panels for gels and western blots are appropriately described as loading on sample processing controls
 - all images in the paper are checked for duplication of panels and for splicing of gel lanes.
- Finally, please ensure that you retain unprocessed data and metadata files after publication, ideally archiving data in perpetuity, as these may be requested during the peer review and production process or after publication if any issues arise.

We hope to receive your revised manuscript within three months. Please get in touch if you think you might need more time.

Please do not hesitate to contact me if you have any questions or would like to discuss these revisions further. We look forward to seeing the revised manuscript and thank you for the opportunity to review your work.

Best regards,

Ben Abbott, PhD
Associate Editor
Communications Medicine
Nature Research

REVIEWER COMMENTS

OVERAL RESPONSE:

Reviewer #1: This is an informative paper describing 'cascade of care' (CoC) outcomes in several higher risk populations in Madrid hospitals. The premise of the paper is important: to document progress in addressing HCV infection given WHO goals of elimination as a public health imperative. The paper provides important view of potential gaps in care and opportunities for intervention. The study population and setting is relevant as the hospital setting is an important place to screen, diagnose and initiate treatment. Overall the methods are well described. I appreciate the well described process of data quality review. The following specific comments if addressed have potential to improve the paper.

1. Abstract: no comments

2. Introduction: relevant to a comment later in the Discussion, if these hospitals or the Ministry of Health in Madrid has a specific model or guidance, it would be good to state that here.

RESPONSE: Thank you for your input. We have now specified in the discussion that the centres participating in this study follow the European Association for the Study of the Liver's HCV practice guidelines, Spanish national HCV strategy, HCV elimination position statement of the Spanish Association for the Study of the Liver and Madrid's HCV elimination strategy to guide HCV care and some detail about these (p 11).

3. Methods:

- page 8: The supp. table is very informative and if the journal allows would be helpful to include in the full article.

RESPONSE: Thank you for your input. The journal did indeed allow for the inclusion of this table in the article. We have included it as 'Table 2' in the manuscript (p 8, 20).

-the FIGURE should better reflect the numbers in the table. I think that the figure does not accurately reflect forward movement on the CoC. This figure has 'running denominator - so the denominator for the bar in sVR treated is from the preceding group (ever treated). Too really examine forward movement in the CoC and missed opportunities for care, a fixed denominator should be used. At the very least using the N that are RNA positive. Using a fixed denominator will be much more informative about lost opportunities with these patients.

RESPONSE: Thank you for your comment. We see your point in terms of how visually the figure may not seem reflective of the forward movement of the cascade of care (CoC). That being said, the figure is indeed reflective of Table 2 and the CoC and we thus propose a different approach from what you suggested to clarify and improve the figure, so that it becomes more clearly reflective of both the table and the CoC. Instead of changing any values, we have added notes under the figure (similar to the ones that we already had under Table 2) to clarify what the denominator is for each point in the CoC (p 23). For instance, for the point you mentioned, 'SVR12 reached', the denominator is equal to those who completed treatment for each corresponding group. We think that using 'HCV-RNA positive' as a denominator instead would be misleading in terms of lost opportunities, since not all HCV-RNA positive patients were eligible for treatment and thus treated. We have also added the categories 'Eligible for treatment' and 'LTFU/unknown SVR12' to the figure, to help in improving the flow of the figure, while demonstrating lost opportunities in a more clear and visual format and staying reflective of the table and the CoC (p 23). Furthermore, we have also explained in the notes under the figure that 'Ever treated' includes those with ongoing and completed treatment, to improve clarity (p 23).

4. Discussion

- Page 11, line 239. The authors haven't said what this Madrid 'model' is. Is there a policy or guideline document that can be referenced?

RESPONSE: Thank you for your comment. As stated above, we have now specified in the discussion that the centres participating in this study follow the European Association for the Study of the Liver's HCV practice guidelines, Spanish national HCV strategy, HCV elimination position statement of the Spanish Association for the Study of the Liver and Madrid's HCV elimination strategy to guide HCV care some detail about them (p 11). We have also edited the

sentence where we mention that due to its success, Madrid could serve as a model to follow for other European cities for HCV elimination, to increase its clarity (p 12).

- Page 11, line 245. Why is this surprising? Is there an active surveillance system? Is liver disease not monitored for in older patients? This statement seems to suggest that more liver disease monitoring is needed in PWID - however - they did not have the highest proportion of liver fibrosis. I think this section needs a bit more of a 'microscope' to reflect what the data show.

RESPONSE: Thank you for this comment. We had already stated that this finding is surprising given that access to DAAs has been universal in Spain since 2017 but have now also edited the text to note that transplant and ALD patients should be better monitored as our findings show they had the highest rates of stages 3-4 fibrosis at diagnosis and pre-treatment initiation, respectively (p 12).

- Page 11, line 246. This recommendation to better focus on PWID is certainly evident in the screening and testing data - this is your highest risk population but has a very low test uptake! Emphasizing this would improve this section.

RESPONSE: Thank you for your suggestion. We edited the text around PWID by adding that they, "are also a very high-risk population group that require additional surveillance and linkage to care according to our screening and testing data. In addition to being one of the highest at-risk groups for HCV globally¹⁴ and late presentation to care in Spain¹⁵, ..." (p 12).

We also wanted the reviewer to know that the data collection ending period was edited from March to May 2021 as we decided, while working on the peer review responses, that even though all raw data came in by March of 2021, during the data verification process, which went on until May, data were edited and so we wanted the time period stated to be reflective of this.

Reviewer #2: In this analysis the authors pretend to describe a cascade of care of HCV in 6 public hospitals in Madrid. In my opinion they do not present a cascade of care, since only individuals still in care were described, but a cross-sectional observational study to assess whether high risk individuals for having HCV were routinely tested and subsequently treated. They show that they have been quite successful in doing this routinely. In the title and methodology to me more fits a microelimination strategy in a public hospital setting aimed at high risk individuals.

The methodology used however remains unclear and should be clarified. As stated it seems to be a cross-sectional micro-elimination program, for which their own physicians collected data but several times in the manuscript it seems that tests were actively ordered for which participants gave their informed consent (as stated in the methods). In other parts the results seem more descriptive and also the discussion suggests a descriptive nature of their work/retrospective data collection.

This should be clarified in the methods and reflected in the title.

RESPONSE: Thank you for your comment. We agree that this is a cross-sectional observational study (the word "observational" was added to the methods in p 4, after "cross-sectional" to ensure clarity) and we also think that this work is indeed reflective of a cascade of care analysis, since by definition this process involves evaluating patient retention across the sequential care stages needed to attain a successful treatment outcome, which is what we have done.

Furthermore, we do not agree that this fits a micro-elimination programme, since it is a retrospective registry review. We have added the word “retrospective” before “registry review” in the methods section (p 4) to ensure clarity, as per your suggestion. In terms of the tests that were ordered, this was all done previous to data collection, and we have edited the text in p 7 to help clarify this.

In the method section a power calculation is presented. To me it is completely unclear why this presented given the observational nature of their work.

RESPONSE: The power calculation is needed to ensure the generalisability of the study findings. We have added some text on p 5 to help clarify this.

Figure 1 describes their key findings. For clarity I would suggest to cluster the different risk groups instead of different steps.

RESPONSE: Thank you for your comment. Since this is a cascade of care (CoC) analysis, we clustered each sequential step in the CoC per high-risk group (except pre-dialysis patients for the reasons explained under the figure), in order to visually reflect the flow of the CoC. As per reviewer #1’s suggestion, we have improved the figure by adding the categories ‘Eligible for treatment’ and ‘LTFU/unknown SVR12’, to help in improving the flow and its clarity by explaining what the denominator is for each point in the CoC and that the category ‘Ever treated’ includes those with ongoing and completed treatment (p 23).

Data was collected by the patients own physicians. Was any form of data verification performed?

RESPONSE: Thank you for your question. As we have now clarified in the methods (p 4), as above, this was a retrospective registry review and as such we have edited the writing with regards to the data collection process on p 5-6, to make things clearer. In terms of the data verification process, we have added and edited some text on p 6 to clarify how this was done.

I assume patients were treated at the hepatology and/or ID departments for HCV. Was treatment and fibrosis data also primary collected by their own physician or by the hepatologist/ID physician?

RESPONSE: Thank you for your question. Your assumption is correct and patients were indeed treated for HCV by the hepatology/ID departments. Treatment and fibrosis data in the registry was collected by the treating physician (i.e., in the hepatology/ID departments). We have added a sentence on this in p 5 to ensure that this is clear.

Were all individuals in care from the different risk groups included? And if yes how was this verified?

RESPONSE: Thank you for your question. We have added some text to the methods section (p 5) to clarify that not all individuals from the different risk groups were included. As explained by the added text, we asked each hospital to provide at least 125 patients per risk group and even when they were unable to reach this number, we were still able to obtain a sample size much larger (n=3,994) than the required sample size (n=546) to ensure generalisability of study findings.

We also wanted the reviewer to know that the data collection ending period was edited from March to May 2021 as we decided, while working on the peer review responses, that even though all raw data came in by March of 2021, during the data verification process, which went on until May, data were edited and so we wanted the time period stated to be reflective of this.

Reviewer #3: The messages here are potentially important ones in the elimination effort for HCV but I am uncertain over a few areas. The major one is how the cohorts were compiled and how complete. I would accept that haemodialysis is a pretty concrete cohort but PWID in a secondary care sector? How do you know and what were they actually in hospital with? Without certainty over the denominator it is difficult to interpret the proportion positive. I would wish to see a more detailed explanation of how cohorts were accumulated and what safeguards there were in place to ensure that whatever search criteria used were universally employed.

RESPONSE: Thank you for your input. We have added some text in the methods section (p 5) to clarify that we asked each hospital to provide at least 125 patients per risk group to reach a sample size large enough to ensure generalisability of study findings and that even when they were unable to do so, we were still able to obtain a sample size much larger (n=3,994) than the required sample size (n=546) for generalisability purposes.

In terms of your point about PWID, we had already stated in the methods section that we worked with addiction clinics associated to the participating hospitals and have now added that it is from these clinics that we obtained the data on PWID (p 5), to improve clarity around this.

As for the safeguards used to ensure universal application of search criteria, as already mentioned on the methods section (p 4), data collection had a set date range (i.e., from September 2019 to May 2021—we edited the latter from March, please see note* below for an explanation) and we have added the exact dates (i.e., 1 September 2019 to 28 May 2021) to improve clarity (p 4). As for patient inclusion criteria, we had already stated and have further edited the text in the methods section (p 4) that this was a “registry review of adult patients (18 years or older) in haemodialysis or pre-dialysis programmes, co-infected with HIV, with advanced liver disease (ALD), with hereditary haematological diseases (HHD), and with transplants, and of people who inject drugs (PWID)” to improve clarity. In terms of the data verification process, we added and edited some text on p 6 to clarify how this was done.

* The data collection ending period was edited from March to May 2021 as we decided, while working on the peer review responses, that even though all raw data came in by March of 2021, during the data verification process, which went on until May, data were edited and so we wanted the time period stated to be reflective of this.

Given this cohort is 2019 to 2021 I find it extraordinary that some well documented high risk groups (haemodialysis, transplant and HIV positive patients) have such high positive rates. In UK practice these were almost the first micro-elimination cohorts and there will no longer be a single eligible HCV positive dialysis patient who is untreated and this has been the case for a number of years. Given policy in Spain is no different it is clear that some additional explanation is needed here and possibly a major call for change.

RESPONSE: Thank you for highlighting this point. As per your suggestion, we have now noted the finding that a large portion of the cohort tested positive for HCV in the discussion (p 12) given that, as we had already stated on p 12, there has been prioritisation of micro-elimination efforts in

these high-risk populations since 2017. This goes along with the text we already had in this section and edited slightly, about the ongoing late diagnosis and presentation to care issue in Spain, to explain this and how this “...stresses the need for better screening efforts towards certain patient populations...” like transplant patients (p 12), a major call for change.